# Performance Evaluation of the STANDARD^TM^ Q COVID-19 and Panbio^TM^ COVID-19 Antigen Tests in Detecting SARS-CoV-2 during High Transmission Period in Mozambique

**DOI:** 10.3390/diagnostics12020475

**Published:** 2022-02-12

**Authors:** Nádia Sitoe, Júlia Sambo, Neuza Nguenha, Jorfelia Chilaule, Imelda Chelene, Osvaldo Loquiha, Chishamiso Mudenyanga, Sofia Viegas, Jane Cunningham, Ilesh Jani

**Affiliations:** 1Instituto Nacional de Saúde, Marracuene 3943, Mozambique; julia.sambo@ins.gov.mz (J.S.); neuza.nguenha@ins.gov.mz (N.N.); jorfelia.chilaule@ins.gov.mz (J.C.); imelda.chelene@ins.gov.mz (I.C.); sofia.viegas@ins.gov.mz (S.V.); ilesh.jani@ins.gov.mz (I.J.); 2Clinton Health Access Initiative, Maputo City 592, Mozambique; oloquiha@clintonhealthaccess.org (O.L.); cmudenyanga@clintonhealthaccess.org (C.M.); 3Global Malaria Program, World Health Organization, 1211 Geneva, Switzerland; cunninghamj@who.int

**Keywords:** antigen rapid test, SARS-CoV-2, clinical evaluation

## Abstract

(1) Background: Laboratory-based molecular assays are the gold standard to detect SARS-CoV-2. In resource-limited settings, the implementation of these assays has been hampered by operational challenges and long turnaround times. Rapid antigen detection tests are an attractive alternative. Our aim is to evaluate the clinical performance of two SARS-CoV-2 rapid antigen tests during a high transmission period. (2) Methods: A total of 1277 patients seeking SARS-CoV-2 diagnosis were enrolled at four health facilities. Nasopharyngeal swabs for rapid antigen and real time PCR testing were collected for each patient. Sensitivity, specificity, positive and negative predictive values, misclassification rate, and agreement were determined. (3) Results: The overall sensitivity of Panbio COVID-19 was 41.3% (95% CI: 34.6–48.4%) and the specificity was 98.2% (95% CI: 96.2–99.3%). The Standard Q had an overall sensitivity and specificity of 45.0% (95% CI: 39.9–50.2%) and 97.6% (95% CI: 95.3–99.0%), respectively. The positive predictive value of a positive test was 93.3% and 95.4% for the Panbio and Standard Q Ag-RDTs, respectively. A higher sensitivity of 43.2% and 49.4% was observed in symptomatic cases for the Panbio and Standard Q Ag-RDTs, respectively. (4) Conclusions: Despite the overall low sensitivity, the two evaluated rapid tests are useful to improve the diagnosis of symptomatic SARS-CoV-2 infections during high transmission periods.

## 1. Introduction

Since the beginning of the COVID-19 pandemic, access to quality diagnosis has been key in timely clinical management of cases and disease control [1]. However, in low-income settings, access to laboratory-based diagnostic testing, including real-time reverse transcription polymerase chain reaction (RT-PCR), has been limited [2] due to lack of adequate infrastructure, complexity of equipment, challenges of sample referral system [3], shortage of reagents, and scarcity of trained personnel [4,5]. These factors often lead to long result turnaround times (TAT), which negatively impact the efficiency of infection control, especially during high transmission periods.

Point-of-care, rapid tests based on antigen detection (Ag-RDT) enable early identification of SARS-CoV-2 infection by obtaining results in 15 to 30 min [6,7], and allow for prompt clinical decision making. In turn, this reduces the demand and pressure on laboratory-based diagnostic testing. However, antigen-detection assays may have lower sensitivity when compared to laboratory-based tests that rely on the amplification of viral nucleic acid. To date, there have been few evaluations of the added value of SARS-CoV-2 Ag-RDT under field conditions in tropical settings. It is expected that specific circumstances, such as environmental conditions, COVID-19 epidemiological context, and prevalence of other relevant pathogens, among others can influence the performance of rapid tests [8,9,10,11,12,13].

We conducted an evaluation of two SARS-CoV-2 Ag-RDTs, Panbio^TM^ and STANDARD^TM^ Q, compared with the gold standard RT-PCR, in real life conditions during a high transmission period in Mozambique.

## 2. Materials and Methods

### 2.1. Study Setting and Participants

A cross-sectional prospective study was conducted during the second wave of high transmission in Mozambique, from January to March 2021, at four health facilities: Hospital Provincial da Matola (HPM) and Centro de Saúde de Marracuene (CSM) in Maputo Province, and Hospital Geral de Chamanculo (HGC) and Hospital Geral de Mavalane (HGM) in Maputo City. Participants aged 18 years old and above, suspected of SARS-CoV-2 infection, and contacts of confirmed cases were enrolled in the study. Recruitment into the study was based on suspected cases presented at the triage rooms. The cases and contacts were defined based on WHO guidelines [14].

For each participant, two nasopharyngeal samples were collected for testing: one was used for the Ag-RDT under evaluation and performed at the health facility of collection, while the second was shipped to the Instituto Nacional de Saúde for the RT-PCR. Only the results of the RT-PCR were issued to the participants. Sample collection and Ag RDT testing was conducted by trained and certified non-healthcare workers that had been assessed for competency by certified Trainers of Trainers [15].

### 2.2. Data Collection

Demographic, clinical, and epidemiological data were prospectively collected at each study site using electronic standardized forms on mobile tablets (Samsung, Suwon-si, Korea, model: TAB A 8”). Laboratory technicians that performed the RT-PCR were blinded to the results generated with the Ag-RDT. Only the results obtained in the laboratory platforms were given to the study participants. The Ag-RDT results were only used for the study purposes.

### 2.3. Antigen Rapid Assay testing

Two SARS-CoV-2 Ag-RDTs were evaluated, the Panbio^TM^ COVID-19 Ag rapid test (Abbott, Jena, Germany, Ref: 41FK10 Lot: 41ADF115A) and the STANDARD^TM^ Q COVID-19 Ag test (SD Biosensor, Suwon-si, South Korea, Ref: Q-NCOV-01G Lot: QCO3020169I), both storable at 2–30 °C. A nasopharyngeal swab with a specimen was inserted into an extraction buffer tube provided by the manufacturer, stirred the swabs for at least five times, and removed it while squeezing the sides to extract liquid from the swab. Subsequently, three to five drops were applied to the specimen well of the test device and the test result was read in 15 to 30 min. Only the results with a valid control line were considered for data analysis. The results without a control line but with a test line were considered invalid. The results without control and test lines were excluded.

### 2.4. Real Time PCR Testing

The nasopharyngeal swabs collected by the study staff were placed into sterile tubes containing 3 mL of viral transport media (iClean, Shenzhen, China, REF. CY-F005-20) and sent to the Instituto Nacional de Saúde on the same day of collection. At the laboratory, samples were stored at 2–8 °C for up to 24 h. Automatic SARS-CoV-2 RNA extraction and amplification was performed on the Abbott m2000 platforms (Abbott Molecular, Taipei City, Taiwan). The assay uses two sets of primers to amplify regions within the highly conserved RNA-dependent RNA polymerase (RdRp) and N genes. The automated Cobas 6800 instrument (Roche, Amadora, Portugal) that detects two genes, E gene and ORF 1ab, was used as back up for when the Abbott m2000 was out of service. In the positive results, two genes were detected. The results with only one gene detected were considered indeterminate. The Ct value was used to determine the SARS-CoV-2 viral load in the positive cases.

### 2.5. Statistical Methods

Two-way contingency tables were used to summarize the data and performance of the two SARS-CoV-2 Ag-RDTs compared with the gold standard. The test performances were assessed by determining their sensitivity, specificity, positive and negative predictive values, and misclassification [16]. The positive and negative predictive values were determined taking the prevalence of the RT-PCR into account. 

The overall observed agreement was given by the Cohen’s kappa coefficient [17]. The statistical analysis was performed using Microsoft Excel 360 (Microsoft Co., Redmond, WA, USA), and the level of significance adopted was 0.05.

## 3. Results

### 3.1. Patient Population

A total of 1277 participants were enrolled between January and March 2021 at four health facilities: CSM (*n* = 298), HGM (*n* = 179), HGC (*n* = 372), and HPM (*n* = 428). Of these participants, 551 and 726 were used for the evaluation of the Panbio and Standard Q, respectively (Figure 1). In view of the high transmission at the time of the study, 209 (37.93%) and 371 (51.10%) individuals were RT-PCR positives, respectively. Most of the participants had Ct values below 25, 92.2% and 89.2% for those tested with the Panbio and Standard Q, respectively. The median age of participants was 38 years (range: 18–88 years old) and 54.3% (694/1277) were female. Most of the participants reported symptoms onset 7 days before presentation (76.0%) (Table 1).

### 3.2. Diagnostic Performance of Panbio^TM^ COVID-19 Ag Rapid Test

The overall sensitivity and specificity of the Panbio Ag-RDT were 41.3% (95% CI: 34.6–48.4%) and 98.2% (95% CI: 96.2–99.3%), respectively. Higher sensitivity was observed in 43.2% (95% CI: 36.0–50.5%) of symptomatic patients when compared to 22.2% (95% CI: 6.4–47.6%) in asymptomatic participants. A sensitivity of 53.3% (95% CI: 42.6–63.7%) and 49.6% (95% CI: 41.1–58.2%) was observed in those with symptoms onset 5 and 7 days before presentation, respectively. The proportion of participants erroneously diagnosed as negative or positive was 19.0% (95% CI: 14.3–24.5%) and 20.8% (95% CI: 16.7–25.3%) in those with onset of symptoms 5 and 7 days before presentation, respectively (Table 2). In patients with symptoms onset for longer than 7 days before presentation, the sensitivity of the test was lower at 25.5% (95% CI: 14.3–39.6%), with 33.9% (95% CI: 25.3–43.3%) of the results being misclassified. The overall positive predictive value (PPV) was 93.3% (95% CI: 88.8–97.8%) and the negative predictive value (NPV) was 52.9% (95% CI: 48.6–57.1%). 

The Cohen’s k among those with symptoms onset 5 and 7 days before presentation was 0.55 (95% CI: 0.53–0.57) and 0.52 (95% CI: 0.51–0.53), respectively. In individuals without symptoms and those with symptoms onset for longer than 7 days before presentation, the Cohen’s k was 0.29 (95% CI: 0.04–0.56) and 0.25 (95% CI: 0.19–0.30), respectively.

The overall agreement between the results yielded by the RT-PCR and the Panbio Ag-RDT was good in subjects with higher viral loads, as depicted by Ct values below 16 (Table 3). In patients with lower viral loads, the overall agreement dropped to 52.7% and 11.9% for those with Ct values below 25 or higher than 26, respectively.

A total of six false positives were observed during the evaluation of the Panbio Ag-RDT. In those cases, all were symptomatic, five with symptoms onset for 7 days before presentation and three contacts of positive confirmed cases.

### 3.3. Diagnostic Performance of STANDARD^TM^ Q COVID-19 Ag Test

The overall sensitivity and specificity of the Standard Q Ag-RDT were 45.0% (95% CI: 39.9–50.2%) and 97.6% (95% CI: 95.3–99.0%), respectively. Higher sensitivity was observed in 49.4% (95% CI: 43.8–55.0%) of symptomatic participants when compared to 13.3% (95% CI: 5.1–26.8%) of asymptomatic participants. A sensitivity of 61.9% (95% CI: 53.9–69.4%) and 54.4% (95% CI: 47.9–61.0%) was observed in those with symptoms onset 5 and 7 days before presentation, respectively. The proportion of participants erroneously diagnosed as negative or positive was 21.0% (95% CI: 16.6–26.0%) and 25.2% (95% CI: 21.2–29.5%) in those with onset of symptoms 5 and 7 days before presentation, respectively (Table 2). In patients with symptoms onset for longer than 7 days before presentation, the sensitivity of the test was lower at 35.2% (95% CI: 25.3–46.1%), with 44.4% (95% CI: 35.8–53.2%) of the results being misclassified. The overall positive predictive value (PPV) was 95.4% (95% CI: 92.3–98.5%) and the negative predictive value (NPV) was 66.4% (95% CI: 62.3–70.4%).

The Cohen’s k among those with symptoms onset 5 and 7 days before presentation was 0.58 (95% CI: 0.57–0.59) and 0.51 (95% CI: 0.49–0.52), respectively. In individuals without symptoms and those with symptoms onset for longer than 7 days before presentation, the Cohen’s k was 0.14 (95% CI: 0.05–0.22) and 0.24 (95% CI: 0.21–0.27), respectively.

The overall agreement between results yielded by the RT-PCR and the Standard Q Ag-RDT was good in subjects with higher viral loads as depicted by Ct values below 16 (Table 3). In patients with lower viral loads, the overall agreement dropped to 55.4% and 12.1% for those with Ct values below 25 or higher than 26, respectively.

A total of eight false positives were observed during the evaluation of the Standard Q Ag-RDT. In those cases, 75.0% were symptomatic, four with symptoms onset for 7 days before presentation and two contacts of positive confirmed cases. 

## 4. Discussion

Timely diagnosis remains a priority in the efforts to control the spread of SARS-CoV-2. Access to laboratory-based PCR testing is challenged by the high costs of molecular tests, the complexity of laboratory-based testing procedures, and the prolonged turnaround time of results. During high transmission periods of COVID-19, positive cases have to be promptly detected for rapid clinical decision making and outbreak control [6,18].

The low-cost, ease of use, and short turnaround time of SARS-CoV-2 Ag-RDTs make them an attractive alternative to laboratory-based molecular tests [6]. Effectively, SARS-CoV-2 Ag-RDTs can contribute to improved diagnostic coverage and reduced burden of specimens’ referral logistics. 

Several studies have described the performance of Ag-RDTs [9,10,11,12,13,19]. Our results demonstrated that the two Ag-RDTs, Panbio COVID-19 Ag and Standard Q COVID-19 Ag, had an overall low sensitivity. Nevertheless, the sensitivity was higher in patients who reported symptoms, especially in those with symptoms onset five or seven days before testing. Lower sensitivity (30.2% to 58.1%) has been observed in different studies. For example, Ristić et al. (2021), found an overall sensitivity of 58.1% (95% CI: 42.1–73.0), which was higher in the first five days following symptoms onset (100%, 83.3%, 66.7%, 71.4%, and 100%, for days 1 to 5, respectively) than in later days (55.6% between the 6th and 10th day) [20]. Additionally, the results obtained from Cohen’s kappa coefficient analysis showed better agreement in symptomatic patients, with symptoms 1 to 5 days before the test, thus reinforcing better applicability of both RDTs in this group of patients [21]. Another study, performed by Mboma et al. (2021), found an overall sensitivity of 30.4% (95%CI: 18.8–90.9%) in adults attending a hospital being higher at 52.9% in symptomatic cases [22]. Furthermore, Jegerlehner et al. (2021), in a different study, found an overall sensitivity of 65.3% (95% CI: 56.8–73.1) and 44.4% (95% CI 24.4–65.1) in asymptomatic cases [23]. The relatively low sensitivity observed in our study for both Ag-RDTs could be attributed to several factors, including deficient specimen collection [10], test performance, and patients’ clinical characteristics, including their viral loads [16]. Our study also observed a lower sensitivity (<35%) in asymptomatic cases, similar to those observed in other reports [22,24], that are probably associated with lower viral loads in those patients [4,20]. Evaluations in Europe and America observed a higher sensitivity of Ag-RDTs when compared to studies in Asia and Africa. Many RDTs are manufactured in Europe and America, which may affect the test performance in Asia and Africa after repeated freeze–thaw procedures during transportation [19]. 

Several new approaches can be used to overcome the low sensitivity of the Ag-RDT in an effort to scale-up the use of these tests for detecting, monitoring, and controlling outbreaks [24]. One, proposed to be used in low prevalence contexts, is the combined use with the RT-PCR in diagnostic algorithms [25]. Another approach, the clinical prediction rules [25], fits in high prevalence scenarios where, in absence of PCR for confirmation, symptomatic patients with negative Ag-RDT results can be clinically predicted as having COVID-19. Considering our Cohen’s kappa results in the asymptomatic patients and those with onset symptoms > 7 days, an extra caution might be required when the clinician tests these patients. Therefore, the use of the RT-PCR should be recommended.

A total of 14 false positive cases were observed on both rapid assays, with no de-mographical or clinical patterns identified. During the use of Ag-RDT for SARS-CoV-2, false positive cases can occur mostly when assay procedures are not followed correctly, when multiple specimens processing in batch mode affects the incubation times for each sample, or through cross-contamination between specimens [26]. Moreover, the time since symptoms onset and swab type can lead to false-negative results in the RT-PCR. For example, Wikramatatna et al. (2020), observed that those with symptoms onset 10 days before testing had a 25% chance of being false-negative using nasopharyngeal swabs and a 47% chance using oropharyngeal swabs [27]. If clinical suspicion of COVID-19 is high, then interpretation of RT-PCR negative results need to carefully consider the epidemiological context [28]. Most of our false positives in the Ag-RDT occurred in symptomatic individuals.

In the context of high transmission of a communicable disease, it is important that diagnostic tests with a high PPV are used. This allows for both clinical and public health decisions to be made with greater confidence. We found PPVs higher than 90% in our study population, especially among patients with symptoms. Thus, in our setting, during a high transmission period, at least 90% of those testing positive on Ag-RDTs were correctly identified as being infected with SARS-CoV-2. In many resource-limited settings, laboratory-based molecular diagnosis is not a viable alternative for efficiently detecting individuals infected with SARS-CoV-2. In these settings, Ag-RDTs are frequently the only option [29], even when or if they have a lower sensitivity. Their utilization, especially in high transmission periods, can effectively contribute to the identification of a high number of infected people, who would otherwise remain undiagnosed. For example, in a hypothetical population of 10,000 people with a positivity rate of 20%, the two evaluated Ag-RDTs would identify 826 and 900 positive cases for the Panbio and Standard Q, respectively.

## 5. Conclusions

Our results show that the evaluated rapid tests have great utility for SARS-CoV-2 diagnosis among symptomatic patients during high transmission periods. Laboratory-based molecular diagnosis with good sensitivity, as RT-PCR, will still be required for SARS-CoV-2 triage among contacts of positive cases, asymptomatic cases, and symptomatic patients with negative results in Ag-RDTs. Training and regular operator competency assessment must be considered for the successful implementation of Ag-RDTs in resource-limited settings.

## Figures and Tables

**Figure 1 diagnostics-12-00475-f001:**
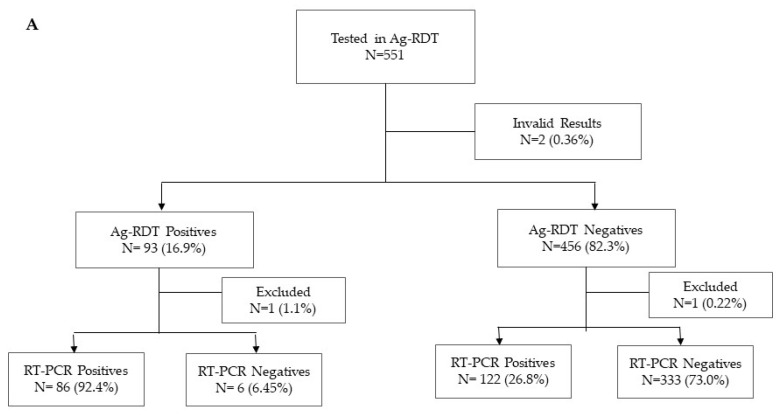
Flow diagram of study participants tested by RT-PCR and Panbio^TM^ COVID-19 Ag rapid test (**A**) and STANDARD Q COVID-19 Ag Test (**B**).

**Table 1 diagnostics-12-00475-t001:** Characteristics of study participants.

		Total		Panbio COVID-19 Ag		Standard Q COVID-19 Ag	
		N = 1277	% Total	N = 551	% Total	N = 726	% Total
Ag-TDR	Negative	1002	78.47	456	82.76	546	75.21
	Positive	273	21.38	93	16.88	180	24.79
	Invalid	2	0.16	2	0.36	0	0
RT-PCR	Negative	674	52.78	340	61.71	334	46.01
	Positive	580	45.41	209	37.93	371	51.10
	Excluded	20	1.57	2	0.36	18	2.48
	Indeterminate	3	0.23	0	0	3	0.41
Health Facilities	CS de Marracuene	298	23.33	19	3.45	279	38.43
	HG Mavalane	179	14.01	111	20.15	68	9.37
	HG Chamanculo	372	29.13	237	43.01	135	18.6
	HP Matola	428	33.51	184	33.39	244	33.61
Age	Median in years	38.0	NA	42	NA	38.0	NA
Sex	Female	694	54.35	303	54.99	391	53.86
	Male	583	45.65	248	45.01	335	46.14
Symptoms	Yes	1075	84.18	480	87.11	595	81.96
	No	202	15.82	71	12.89	131	18.04
Onset symptoms	≤7 days	817	76.00	364	75.83	453	76.13
	>7 days	252	23.44	116	24.17	136	22.86
	No information	6	0.56	0	0	6	1.01

**Table 2 diagnostics-12-00475-t002:** Analytic performance of two antigen rapid assays for SARS-CoV-2 detection.

		Sensitivity	Specificity	Positive Pred. Value	Negative Pred. Value	Misclassification	Observed Agreement	Chance Agreement	Cohen’s Kappa
Panbio COVID-19 Ag	Overall	41.3% (34.6–48.4%)	98.2% (96.2–99.3%)	93.3% (88.8–97.8%)	52.9% (48.6–57.1%)	23.4% (19.9–27.2%)	76.6% (31.9–80.1%)	56.7%	0.45 (0.44–0.46)
	With symptoms	43.2% (36.0–50.5%)	97.9% (95.5–99.2%)	93.0% (88.4–97.7%)	55.3% (50.8–59.8%)	23.9% (20.2–28.0%)	76.1% (32.0–79.8%)	55.4%	0.46 (0.45–0.47)
	Without symptoms	22.2% (6.4–47.6%)	100.0% (93.3–NA)	100.0% (100.0–NA)	30.4% (19.4–41.4%)	19.7% (11.2–30.9%)	80.3% (25.4–88.8%)	71.9%	0.29 (0.04–0.56)
	Onset symptoms ≤ 5 d ^1^	53.3% (42.6–63.7%)	97.4% (93.5–99.3%)	91.3% (84.7–98.0%)	56.5% (50.2–62.8%)	19.0% (14.3–24.5%)	81.0% (32.4–85.7%)	56.5%	0.55 (0.53–0.57)
	Onset symptoms ≤ 7 d	49.6% (41.1–58.2%)	97.7% (94.8–99.3%)	93.1% (88.1–98.0%)	57.3% (52.2–62.4%)	20.8% (16.7–25.3%)	79.2% (32.6–83.3%)	55.4%	0.52 (0.51–0.53)
	Onset symptoms > 7 d	25.5% (14.3–39.6%)	98.4% (91.6–100.0%)	92.9% (79.4–100.0%)	50.4% (41.0–59.8%)	33.9% (25.3–43.3%)	66.1% (25.7–74.4%)	54.7%	0.25 (0.19–0.30)
Standard Q COVID-19 Ag	Overall	45.0% (39.9–50.2%)	97.6% (95.3–99.0%)	95.4% (92.3–98.5%)	66.4% (62.3–70.4%)	30.2% (26.7–33.6%)	69.8% (33.8–73.3%)	48.6%	0.41 (0.41–0.42)
	With symptoms	49.4% (43.8–55.0%)	97.6% (94.8–99.1%)	96.4% (93.6–99.2%)	71.5% (67.2–75.9%)	29.8% (26.0–33.6%)	70.2% (35.1–74.0%)	47.2%	0.44 (0.43–0.44)
	Without symptoms	13.3% (5.1–26.8%)	97.6% (91.7–99.7%)	75.0% (45.0–100.0%)	37.6% (29.0–46.3%)	31.8% (23.9–40.6%)	68.2% (24.8–76.1%)	63.2%	0.14 (0.05–0.22)
	Onset symptoms ≤ 5 d	61.9% (53.9–69.4%)	97.9% (94.1–99.6%)	97.1% (93.8–100.0%)	73.8% (67.8–79.9%)	21.0% (16.6–26.0%)	78.8% (36.9–83.4%)	49.1%	0.58 (0.57–0.59)
	Onset symptoms ≤ 7 d	54.5% (47.9–61.0%)	98.0% (95.1–99.5%)	96.9% (94.0–99.9%)	71.1% (66.0–76.2%)	25.2% (21.2–29.5%)	74.8% (35.7–78.8%)	48.6%	0.51 (0.49–0.52)
	Onset symptoms > 7 d	35.2% (25.3–46.1%)	95.6% (84.9–99.5%)	93.9% (85.8–100.0%)	74.3% (65.7–82.8%)	44.4% (35.8–53.2%)	55.6% (27.6–64.2%)	41.9%	0.24 (0.21–0.27)

^1^ d = days.

**Table 3 diagnostics-12-00475-t003:** Agreement between RT-PCR positives and Ag-RDT regarding to CT values.

		≤15	≤25	≥26
Panbio COVID-19 Ag	Overall	90.8% (59/65)	52.7% (77/146)	11.9% (7/59)
	With symptoms	90.3% (56/62)	52.9% (73/138)	14.3% (7/49)
	Without symptoms	100% (3/3)	50% (4/4)	0% (0/10)
	Onset symptoms ≤ 5 d	90.2% (37/41)	64.6% (42/65)	13.0% (3/23)
	Onset symptoms ≤ 7 d	89.9% (49/51)	61.5% (59/96)	15.8% (6/38)
	Onset symptoms > 7 d	100% (7/7)	35.7% (10/38)	0% (0/10)
Standard Q COVID-19 Ag	Overall	87.0% (109/124)	55.4% (153/276)	12.1% (14/116)
	With symptoms	89.2% (107/120)	60.0% (147/245)	13.9% (14/81)
	Without symptoms	50.0% (2/2)	0% (0/15)	0% (0/15)
	Onset symptoms ≤ 5 d	94.5% (70/74)	80% (93/125)	13.3% (6/45)
	Onset symptoms ≤ 7 d	92.5% (87/94)	66.3% (118/178)	13.4% (9/67)
	Onset symptoms > 7 d	64.7% (11/17)	50% (26/52)	12.1% (4/33)

## Data Availability

The datasets generated during this study are not publicly available due to the limitations of the participants’ consent forms.

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
