# Peer review of "Performance Evaluation of the STANDARDTM Q COVID-19 and PanbioTM COVID-19 Antigen Tests in Detecting SARS-CoV-2 during High Transmission Period in Mozambique"

_diagnostics, 2022, doi:10.3390/diagnostics12020475_

Round 1
Reviewer 1 Report
Summary
The study conducts a comparison of two rapid diagnostic tests for SARS-CoV-2 against the gold standard PCR in clinical use, in four hospitals in Mozambique during a period of high transmission. The authors analyse the sensitivity, specificity and other measures of agreement between these tests and PCR.
Review
Evaluating rapid diagnostic tests as used in real clinical practice is important given their utility, particularly in resource-limited settings where PCR is difficult to conduct in a timely fashion. This study makes a clear comparison of two types of rapid antigen test against the gold standard PCR. Conducting the study during a high transmission period is also of value because it is more likely to demonstrate the efficacy of these tests under operational pressure.
The only conceptual issue I would raise is around the treatment of positive antigen results with negative PCR as false positives. This is appropriate as this is a study vs PCR as gold standard, but there is no mention of the possibility of a failure of PCR despite an actual infection. I think it would be worth acknowledging in the discussion that this is a possibility, especially given the epidemiological evidence of infection (lines 133-135 and 154-156).
Additional comments
- Line 89
"The automated Cobas 6800 instrument (Roche, Portugal) that detects two genes, E gene and ORF 1ab, was used as backup for Abbott m2000, when out of service."- What does this mean? Did it occur frequently? Is there a record?
- If this information is unavailable, this is not a black mark on the study, because breakdown of the equipment and the use of a replacement is not unexpected in a pressurised pandemic situation with limited resources.
- Line 109
"Most of the participants referred symptoms onset 7 days before presentation (76.0%) (Table 1)."- "referred" -> "reported"?
- Figure 1
- Not clear to me the difference between excluded, indeterminate and invalid results
- Table 1
- Horizontal lines to separate the results from Ag-TDR, RT-PCR etc would make it easier to read
- Same comment as for Figure 1: ensure that Invalid, Excluded and Indeterminate are explained somewhere
- Line 174
- Lower sensitivity (30.2% to 58.1%) amongst symptomatic cases has been observed in different studies"
- Where does the 30.2% come from? In the Mboma et al study, they report 30.4% overall, and 52.9% symptomatic.
- If it's referring to one of these, please correct the comparison.
- If it's referring to something else, please clarify.
- Line 193
- What does "no demographical or clinical patterns associated" mean?
Reviewer 2 Report
The manuscript contained several typing errors where words are separated by dashes please correct them.
Reviewer 3 Report
Major comment
As this study use human-derived materials, IRB approval or something equivalent might be required.
The approval of the human-derived materials should be noted in the manuscript.
How did the authors calculate PPV and NPV? Do the authors consider the prevalence of COVID-19 to calculate PPV and NPV?
What was the prevalence of COVID-19? Based on total population? Or based on RT-qPCR results?
This should be clearly described in the text. Without considering prevalence, PPV and NPV can be overexaggerated.
The authors mentioned Cohen’s Kappa coefficient. But there is no mention of the coefficient in the result and discussion. The coefficient between methods should be clearly stated in the text.
The authors only showed the ‘agreements’ but also should calculate the hypothetical probability of chance agreement for the coefficient, especially in the case of low sensitivity.
Minor
There are unnecessary ‘-‘ in the text. For example, ‘la-boratory-based’ should be changed to ‘laboratory-based’
These should be removed.
Reviewer 4 Report
This paper is fairly well written, but has quite a few revisions required to improve the quality of the manuscript.
The entire manuscript has a very large number of text words that are incorrectly hyphenated throughout the manuscript, presumably due to the authors originally trying to add hyphens at the end of text lines. All of these hyphens should be removed.
The Introduction is quite lean and could benefit by more references on the range of different diagnostic methods used for Covid-19 and particularly some more recent references 2021-2022 on this topic.
Also, the Discussion section should include more comparisons of the author's results to those of other published studies on this specific subject. There are many more and recent articles on this available in the literature that could enrich your discussion of results and comparisons of diagnostic methods.
The Conclusions are adequate and supported by results, but some additional comments about the potential applications, significance, and usefulness of this new comparative information would be helpful as far as recommendations with regard to which diagnostic methods are more reliable and useful for accurate Covid-19 diagnoses.
Round 2
Reviewer 3 Report
Most of my previous comments were properly answered.
There are two comments.
I suggest the calculation of PPV and NPV with the consideration of the prevalence. Although PPV and NPV without prevalence are not wrong, considering prevalence might be better. I think the authors can calculate the prevalence from RT-qPCR results as it might be the prevalence in the clinics.
PPV= sensitivity * prevalence/[sensitvity*prevalence+(1-specificity)*(1-prevalence)]
NPV= specificity * prevalence / [specificity*prevalence+(1-sensitivity)*(1-prevalence)]
Considering Cohen's Kappa of asymptomatic and Onest symptom> 7 days, extra caution might be required when the clinician test these patients. This might be stated in the text.
